# Technological Properties of Inulin-Enriched Doughs and Breads, Influence on Short-Term Storage and Glycemic Response

**DOI:** 10.3390/foods13172711

**Published:** 2024-08-27

**Authors:** Michele Canale, Rosalia Sanfilippo, Maria Concetta Strano, Anna Rita Bavaro, Margherita Amenta, Michele Bizzini, Maria Allegra, Sebastiano Blangiforti, Alfio Spina

**Affiliations:** 1Research Centre for Cereal and Industrial Crops, Council for Agricultural Research and Economics (CREA), Corso Savoia, 190, 95024 Acireale, Italy; rosalia.sanfilippo@crea.gov.it (R.S.); alfio.spina@crea.gov.it (A.S.); 2Research Centre for Olive, Fruit and Citrus Crops, Council for Agricultural Research and Economics (CREA), Corso Savoia, 190, 95024 Acireale, Italy; mariaconcetta.strano@crea.gov.it (M.C.S.); maria.allegra@crea.gov.it (M.A.); 3Institute of Sciences of Food Productions (ISPA), National Research Council (CNR), Via G. Amendola, 122/O, 70126 Bari, Italy; annarita.bavaro@ispa.cnr.it; 4Stazione Consorziale Sperimentale di Granicoltura per la Sicilia, Via Sirio, 1, 95041 Caltagirone, Italy; michele.bizzini@gmail.com (M.B.); blangiforti@granicoltura.it (S.B.)

**Keywords:** bread, functional food, glycemic index, inulin, leakage rate, rheological properties, water absorption

## Abstract

The use of inulin in food is highly appreciated by consumers because of its prebiotic effect. In this study, the effects of increasing additions (5, 10 and 20%) of inulin as a substitute for wheat flour in bread production were investigated with regard to the physical, technological and rheological properties of the flour blends. Inulin reduced the water-binding capacity from 1.4 g/100 g with 0 flour to 0.80 g/100 g with the 20% inulin addition, while there were no statistical differences in the oil-binding capacity. The addition of inulin also influenced the yeast rates, especially in the samples with 5 and 10% addition. On the farinograph, inulin caused a reduction in water absorption (40.75 g/100 g with 20% inulin), an increase in dough development time (18.35 min with 10% inulin) and dough stability (13.10 min with 10% inulin). The mixograph showed a longer kneading time for the sample with 20% inulin (8.70 min) than for the control (4.61 min). In addition, there was an increase in dough firmness and tightness due to the addition of inulin (W: 203 × 10^−4^ J; P/L: 4.55 for the 20% inulin sample) compared with the control. The physical and technological properties of the loaves were evaluated at time 0 and after 4 days (T4). The addition of inulin reduced the volume of the bread while increasing the weight, albeit with a weight loss at T4 (compared to T0) of 4.8% for the 20% inulin and 14.7% for the control. The addition of inulin caused a darkening of the crust of the enriched bread, proportional to the increase in inulin content. In addition, the inulin content ranged from 0.82 g/100 g in the control to 14.42 g/100 g in the 20% inulin bread, while the predicted glycemic index ranged from 94.52 in the control to 89.39 in the 20% inulin bread. The available data suggest that the formulation with 5% inulin provides the highest performance.

## 1. Introduction

Wheat is one of the most important food sources in the world due to its high adaptability to different environments and its potential to meet global food needs [1]. Bread is the main food product made from wheat and also one of the most consumed foods in the world. It is rich in carbohydrates and is a staple food for people all over the world.

As an essential part of the daily diet, bread lends itself well to the inclusion of natural ingredients that can improve the nutritional profile of the product and make it healthier. This can be achieved by blending wheat flour with other plant matrices that can provide protein and fiber [2].

In recent years, interest in bread consumption has increased, which is related to the improved nutritional properties due to biofortification with proteins, fiber and other nutrients. Functional products have attracted the attention of the food industry because they contribute to the maintenance of optimal health and well-being [3]. Indeed, the addition of health-promoting compounds, such as prebiotic fiber, to bread is becoming a widespread practice in the development of functional foods [4]. In this sense, prebiotics are non-digestible food ingredients that support the improvement of metabolic activities by optimizing the composition of the gut microbiome and improving host health [5,6]. This is achieved by the ability of prebiotic fiber to selectively stimulate the growth of beneficial microorganisms in the colon while inhibiting the growth of harmful microorganisms. As is known, dietary fiber is classified into ‘insoluble’ and ‘soluble’ [7], the former negatively affecting the technological and rheological properties of bread [8], in contrast to the soluble fiber, which is able to act actively in the gut and reduce glucose absorption, thus lowering postprandial blood glucose level [9]. This could be of great benefit for the prevention and risk reduction of chronic metabolic diseases [10].

Among the negative effects of excessive consumption of soluble fiber, as indicated by Singh et al. [11] in a study related to the occurrence of icteric hepatocellular carcinoma (HCC) in laboratory mice, inulin may increase the risk of HCC occurrence due to dysregulated microbial fermentation. Furthermore, although its important prebiotic action is recognized, its use in the diet of infants and preschool children is strongly discouraged and prohibited in dedicated food formulations.

The strong focus on the use of dietary fiber in bread is due to consumer awareness of its potential against gastrointestinal and cardiovascular diseases [5]. Inulin is one of the dietary fibers suitable for this exact purpose. Inulin is a natural storage fructan polysaccharide obtained from the roots of *Inula helenium*. It has a linear chain of fructose units with glucose units at the reducing end [12]. The main sources are *Cichorium intybus* (chicory) and *Helianthus tuberosus* (Jerusalem artichoke) which are used for food and medicine. Several health benefits are attributed to inulin. The available literature suggests that inulin can protect against oxidative stress and thereby prevent inflammatory responses associated with oxidative stress [13]. As already mentioned, inulin, as a dietary fiber, is able to promote the development of beneficial microorganisms in the gut. In addition, there is evidence in the literature of the benefits of fiber intake in the treatment of gastrointestinal disorders such as constipation and irritable bowel syndrome, as well as in the treatment of obesity and diabetes [14]. It also has hygroscopic properties, so inulin can reduce the water available to starch and affect its gelatinization, resulting in lower carbohydrate absorption and lower glycemic value in foods [15]. On a physiological level, inulin’s ability to bind water means that it can contribute to a better feeling of satiety. Inulin tends to bloat in the stomach when it absorbs water. It is also a low-calorie sweetener, making it potentially suitable as a substitute for sucrose in food formulations, especially in low-calorie or diabetic diets [16].

In the food industry, inulin is extremely versatile and is used for multiple purposes and in a variety of preparations, such as dairy products, ice cream, bread and beverages. It is used as a prebiotic source, as a substitute for sugar as a sweetener, as a substitute for fat and as dietary fiber, as well as to modify texture [17]. It appears in the form of a very fine white powder with a neutral taste, so its use does not affect the taste, smell or general appearance of the product. Indeed, the addition of inulin to baked goods is recommended to provide softness, increase moisture and improve rheological properties, as it acts similarly to fats [18].

This study investigated the influence of adding different amounts of inulin (5, 10, 20%) on the physico-chemical, rheological and technological properties of dough and bread at times 0 and 4 days after baking. The aim of this article is to determine the best range for the addition of inulin and to observe how these different addition levels improve or do not improve the shelf life of bread.

## 2. Materials and Methods

### 2.1. Flour and Inulin

Sicilian bread wheat flour ‘type 0′ was produced by the agricultural cooperative ‘Valle del Dittaino’ a.r.l. in Assoro (Enna, Italy) (latitude 37°57.17 N; longitude 14°44.87 E) and was also used in the production of buns.

Commercial inulin was acquired by ‘Città del gelato a.r.l’ (Noci, Bari, Italy) and produced by Sensus (Oostelijke Havendijk 15, 4704 RA, Roosendaal, The Netherlands). It is a natural powdered food ingredient extracted from chicory roots. This inulin (short chain) from chicory is a polydisperse mixture of linear fructose polymers with mostly a terminal glucose unit coupled by means of β(2-1) bonds. The number of units (degree of polymerization) can vary between 2 and 60.

The composition of dry matter is as follows:−Inulin ≥ 90%;−Fructose, glucose and sucrose ≤ 10%;−The average chain length is 8–13 monomers;−Ash ≤ 0.2%.

Various amounts of inulin were added to the flour: 5%, 10% and 20%.

### 2.2. Technological Tests

#### 2.2.1. Water-Binding Capacity and Oil-Binding Capacity

The flour was tested for water-binding capacity (WBC) and oil-binding capacity (OBC) according to the method described by Sanfilippo et al. [19] An amount of 2 g of flour was added to 24 mL of distilled water or sunflower oil and kept at 20 °C for 60 min with stirring. The samples were centrifuged at 4200 rpm × 30 min (Heraeus Multifuge X3 FR, Thermo Fisher Scientific, Waltham, MA, USA), and the solid residue was weighed. Analyses were performed in triplicate.

#### 2.2.2. Leavening Test

Wheat flour samples with different percentages of inulin and the control flour were mixed to produce doughs that were tested for leavening, according to the method described by Canale et al. [20] The leavening rate of the dough was calculated as follows:$$LR=\frac{\left(V-V_{0}\right)}{V_{0}}\times 100$$
where

V = volume measured after n minutes;V_0_ = initial volume at time 0;The analyses were carried out in triplicate.

### 2.3. Rheological Tests

The dough strength was measured with a mixograph (National Mfg. Co. (Lincoln, NE, USA)) according to the AACC 54-40.02 method [21]. The analyses were carried out in triplicate.

The doughs were also tested with a Brabender farinograph (Duisburg, Germany), equipped with a 300 g mixing chamber, to determine the water absorption capacity, dough development time, dough stability and degree of softening, according to the AACC 54-21 method [21].

An alveograph (Tripette et Renaud, Chopin Technologies, Villeneuve-la-Garenne, France) equipped with the Alveolink software (W1.04/99) (Tripette et Renaud, Chopin Technologies, Villeneuve-la-Garenne, France) was used to determine the dough strength (W) and the tenacity/extensibility ratio (P/L) according to the UNI 10783:1999 [22] by modifying the mixing times for the INU10 and INU20 by 12 and 20 min more, respectively.

The analyses were carried out in triplicate.

### 2.4. Bread Production and Physical Analyses

The breads were made according to the recipe given in Table 1. The other raw materials used were commercial yeast (Lievital, Lesaffre Italia spa, Parma, Italy), salt (Sosalt spa, Trapani, Italy), inulin (‘Città del gelato s.r.l’, Noci, Bari, Italy) and water (by farinograph absorption at 500 B.U.) The dough was mixed in an experimental mixer (National Manufacturing Co., Lincoln, NE, USA) at 25 °C for the time indicated by the dough development on the farinograph (Table 2). It was left to rise in a thermostatic chamber (Giorik, Sedico, Italy) equipped with a steam humidifier (SD/SD series, Carel, Brugine, Italy) at 32–35 °C and 75–80% RH for 90 min. They were then divided, placed in metal baking tins (7 × 18 × 5 cm) and left to rise again for 90 min under the same conditions. Baking took place in an electric oven (Giorik, Sedico, Italy) for 7 min at 215 ± 5 °C, followed by 33 min at 165 ± 5 °C.

The following physical properties were determined on the resulting breads: volume, height, weight, moisture, porosity of the crumb, texture, color of the crumb and the crust. 

The volume of the bread was determined using the rape seed displacement method according to AACC 10-05 [21]. Bread height was measured using a digital caliper (Digi-MaxTM, SciencewareR, Staten Island, NY, USA).

Bread moisture was determined according to the AOAC 935.25 method [23] by drying the bread in a Memmert oven at 103 °C to constant weight. The results were expressed as a percentage of relative humidity (RH%).

The porosity of the crumb was determined by visual comparison of the center slice of each bread with eight Dallmann reference images, representing a cross-section of bread with different crumb structures. Crumb porosity was assessed using the 8-level Mohs scale, modified from Dallmann [24], where 1 indicates an uneven structure (i.e., with large and irregular cells) and 8 indicates a uniform, compact structure (i.e., with small and regular cells) [25]. 

The hardness of the crust of the bread was evaluated using a texture analyzer (Zwick Röell Z 0.5, Ulm, Germany) equipped with a cylindrical stainless steel flat probe with a diameter of 8 mm at a test speed of 1 mm/ and an applied deformation of 20 (force shutdown threshold). The breaking point of the resulting crust was measured in Newton (N). 

The texture of the bread was also tested for the following parameters: elasticity, cohesiveness, gumminess (N) and chewiness (N × mm). The measurements were carried out on slices of bread (15 mm thick) using a TPA (Texture Profile Analysis) test according to the method of Rózylo et al. [26] with slight modifications. A TPA test was performed using a texture analyzer (Zwick Röell Z 0.5, Ulm, Germany) equipped with a stainless-steel compression probe with a diameter of 75 mm, with double compression at a depth of 50% and 10% at a speed of 1 mm/s.

The percentage of water loss during storage (weight loss) was calculated using the following formula:$$\left(\frac{RH_{T0}-RH_{Tx}}{RH_{To}}\right)\cdot 100$$
where

RH_T0_ = bread moisture at T0;RH_TX_ = bread moisture at T2 or T4.

### 2.5. Color Determinations

A CR 200 colorimeter (Minolta, Osaka, Japan) was used for color evaluation according to the CIELab colorimetric model. The results were expressed in terms of L* (for lightness), a* (to indicate the transition from green to red) and b* (to indicate the transition from blue to yellow). The brown index, which indicates the tendency to darken from 0 to 100, was then calculated (100 − L*). ΔE was calculated using the following formula:$$\Delta E=\sqrt{{\left({BI}_{1}-{BI}_{2}\right)}^{2}+{\left(a_{1}-a_{2}\right)}^{2}+{\left(b_{1}-b_{2}\right)}^{2}}$$
where

*BI* = brown index;*a* = red index;*b* = yellow index.

Analyses were carried out in triplicate.

### 2.6. Determination of Bread Staling Rate

The bread was stored for 5 days at 25 °C and packed in paper bags. During storage, the breads were measured for hardness and moisture on the day of baking (T0), then 2 days (T2) and 5 days (T4) after baking to determine the aging rate according to the AACC 10-10.03 method [21]. The analyses were carried out in triplicate.

### 2.7. Inulin Extraction and Quantification

The extraction of inulin and quantification of total fructose after acid hydrolysis were performed on the samples according to the method described by Bavaro et al. [27] with slight modifications. In brief, inulin and free sugars such as glucose, fructose and sucrose were extracted from 1 g of dry bread in 20 mL of water at 100 °C for 1 h. Subsequently, the extracts were centrifuged at 5000× *g* at room temperature for 15 min and filtered at 0.45 μm before quantification of the free sugars by HPLC. Chromatographic separation of sugars (glucose, fructose and sucrose) was performed using a Dionex CarboPac PA1 column and a Carbopac PA1 guard column in isocratic mode with elution of 150 mM NaOH at a flow rate of 1 mL/min. The sugar analyses were performed using a Dionex DX500 HPLC system equipped with a GP50 gradient pump, an ED40 Electrochemical Detector in Pulsed Amperometric Detection (PAD) and the DionexPeaknet 5.11 chromatography software. Simultaneously, an aliquot of each recovered supernatant was hydrolyzed with 0.03N HCl at 70 °C for 30 min. Then, the samples were cooled, filtered and analyzed for quantification of inulin as total released fructose by HPLC-PAD, as previously described.

The amount of fructose from inulin was calculated using the following equation:Fi = Ft − Ff − Fs
where Fi = fructose from inulin, Ft = total fructose after hydrolysis, Ff = free fructose and Fs = fructose from sucrose before hydrolysis.

The inulin content (I) was calculated according to the method described by Steegmans et al. [28] and considering the correction for the glucose part of the inulin and for the loss during hydrolysis:I = 0.995 × Fi

The results were expressed as grams of inulin in 100 g of bread.

### 2.8. Predicted Glycemic Index (pGI) Evaluation

To determine the predicted glycemic index (pGI) of inulin-enriched bread, an in vitro model system was used according to the method proposed by Goñi et al. [29] and described by Garbetta et al. [30]

In brief, bread samples were digested with pepsin (0.1 g/mL) in HCl-KCl buffer (pH 1.5) at 40 °C for 1 h and then with α-amylase (48 U/g sample) in Tris-maleate buffer (pH 6.9) at 37 °C in a shaking water bath. After 0, 30, 60, 90, 120, 150 and 180 min, 1 mL of this solution was removed and incubated at 100 °C for 5 min to inactivate the enzyme. Then, the solutions were cooled and centrifuged at 10,000× *g* at 4 °C and 500 µL of each supernatant was incubated with amyloglucosidase (330 U/mL) in 1.5 mL sodium acetate buffer (pH 4.75) at 60 °C for 45 min. Subsequently, the amount of glucose released was quantified spectrophotometrically at 510 nm using a commercially available enzymatic kit (K-GLUC, Megazyme) based on glucose oxidase/peroxidase (GOPOD) enzyme system. The glucose released during digestion was plotted against time from 0 to 180 min to calculate the area under the curve (AUC). The hydrolysis index (HI) of each bread was calculated as the ratio between the AUC of the samples and the area of white bread used as a reference sample according to the following expression:HI = (AUC sample)/(AUC reference) × 100

An expression was used to calculate pGI as follows:pGI = 39.71 + 0.549 × HI

### 2.9. Statistical Analysis

All data (mean ± standard deviation) were subjected to one-way analysis of variance (ANOVA) using Statgraphics^®^ Centurion XVI (Statpoint Technologies, The Plains, VA, USA) software. The difference between the means was determined using Tukey’s test at a probability level *p* ≤ 0.001 for all the parameters, except for the red index of the crust (*p* ≤ 0.01), inulin content, pGI and the brown index of the crumb (*p* ≤ 0.05).

A principal component analysis (PCA) was carried out on the entire dataset, including the physical and chemical characteristics of flours, doughs and breads with different inulin integrations. PCA was performed using PAST, PAleontological STatistics software package (4.04), 2011 [31].

## 3. Results and Discussion

### 3.1. Water-Binding Capacity and Oil-Binding Capacity

In contrast to other fibrous matrices, the addition of which increases water absorption and retention capacity, inulin behaves in exactly the opposite way.

Figure 1 shows the comparison of the water (WBC) and oil (OBC) uptake of Flour 0 and the three inulin mixtures. There are no statistically significant differences between the samples for OBC, which ranges from 2.29 g/g d.m. (Flour 0) to 2.42 g/g d.m. (INU5). For WBC, the values range from 1.38 g/g d.m. in Flour 0 to 0.80 g/g d.m. in the sample with 20% supplementation, indicating a decrease in absorption capacity with increasing inulin content. This finding was also observed by other authors [32], indicating a significant decrease in WBC with inulin supplementation above 6.5%.

### 3.2. Rheological Data of Flours and Leavening Rate of the Dough

Table 2 shows the physical properties of the pure wheat flour and the inulin supplement. As you can see, the farinographic data show a general deterioration of the dough properties at high inulin concentrations (20%). At higher inulin supplementation, the dough development time increased compared with the control (1.80 min), with the maximum at INU10 (18.35 min). Stability is highest at INU10 (13.10 min) and lowest at INU20 (6.35). Finally, the addition of inulin has an effect on water uptake, which drops from 57.5 g/100 g for the control to 40.8 g/100 g for INU20, confirming the results of the water-binding capacity test (Figure 1). The farinographic data observed are in agreement with the results of other authors [32,33,34] who evaluated the development from 0% to 10% supplementation in wheat flour. The increase in development time and decrease in absorbed water could be due to the interaction of inulin with gluten and starch. In the first case, the inulin tends to absorb water earlier and slow down the formation of the gluten mesh [35]. The decrease in the water-holding capacity of the dough is caused by the inulin molecules, which tend to form a protective barrier around the starch particles and prevent contact between the water molecules and the starch particles. This limits the typical swelling of the starch particles, which is responsible for the increased water-binding capacity. As also observed by other authors [36,37,38], the higher the percentage of inulin present, the lower the water-binding capacity.

The mixograph data, especially the mixing time, confirm what was found in the farinograph. The addition of inulin tends to increase the kneading time (8.70 min, INU20) compared with the control (4.61 min), which is due to the delay in the formation of the gluten tissue.

The addition of inulin in higher concentrations caused an increase in the strength and tightness of the dough, with maximum values in INU20 (W: 203 × 10^−4^ J; P/L: 4.55) compared with the control (W: 154 × 10^−4^ J; P/L: 0.82). This is reflected in the doughs, which change from a looser and stickier structure in the control to a more compact one as the inulin supplementation increases (Figure 2). The increase in strength proportional to the addition of inulin is consistent with other authors [39] who have also reported on the addition of up to 20% of wheat flour.

The addition of inulin alters the fermentative capacity of a dough, mainly as a result of the decrease in water content, as shown in Figure 3.

In detail, different behavior is generally observed between 5–10% and 20% of integration, and the addition of inulin up to 10% supplementation results in a leavening rate of 2.4 times the initial volume, which is higher than the control rate of 1.9 times, in the total 60 min of leavening.

In the case of INU20, the higher fiber content resulted in a shorter leaving time (50 min) and very low growth rates (1 times the initial volume) compared with the other theses. This could be due to two factors: on the one hand, the reduced amount of water needed to form a correct dough affected the correct activity of the yeast. In fact, as observed by several authors [40,41], the reduction of water or the increase of NaCl causes a reduction in the activity of *Saccharomyces cerevisiae* and, consequently, lower consumption of glucose and ethanol, responsible for the fermentation process with CO_2_ production.

The second point is linked to the increase in fiber content, which, as observed by other authors [42], reduces the fermentation capacity and the growth of the dough volume.

### 3.3. Physico-Chemical Characteristics of the Breads for Short Storage

The addition of inulin had a significant effect on the bread, especially at 10 and 20%. As shown in Table 3, the final moisture content at T0 is affected by the lower addition of water added for dough formation as the percentage of inulin addition gradually increases. The values show a significant difference between the samples, ranging from 35.72 g/100 g for the control to 27.06 g/100 g for INU20. This trend also continues at T2 (31.67 g/100 g in the control and 24.73 g/100 g in INU20) and T4 (27.80 g/100 g for the control and 24.32 g/100 g for INU20). Taking into account the differences in water addition to forming the correct dough, the ΔRH was calculated, which showed that the addition of 10% or more inulin led to a significant reduction in moisture loss compared with the control. Specifically, the reduction with INU20 was 4.9% at T2 (after 3 days) and 10.1% at T4 (after 5 days), compared with the control, which had much higher values of 11.3% at T2 and 22.2% at T4.

This improved moisture retention capacity of inulin-enriched bread has also been observed by other authors [36,43,44], with a decrease in water migration capacity from the crumb to the crust, corresponding to a higher moisture content of the bread and the crust itself compared with the control bread.

Some authors worked with an addition level of 3%, and although they did not find significant statistical differences, they confirmed the described behavior [45].

This ability to retain more moisture during baking can be assessed by the change in weight at T0 and in subsequent measurements. At T0, the range was between 153.99 g (control) and 160.92 *g* (INU20), with a weight loss (starting from 180 g of dough) of 10.4% for INU20 and 14.4% for the control. At T2, the range was 142.70 g (control) to 156.19 g (INU20), with a weight loss (compared to T0) of 2.9% for INU20 and 7.3% for the control. Finally, at T4, the range was between 131.32 g (control) and 153.20 g (INU20), with a weight loss (compared to T0) of 4.8% in INU20 and 14.7% in control. In previous studies [20] regarding artichoke flour, containing both insoluble and soluble fiber (inulin), the same trend was observed in durum wheat semolina. The fiber-enriched bread retained a higher moisture content and consequently a higher weight than the control bread made with only flour or semolina.

Regarding volume, as can be seen from Figure 4, the control bread produced a very high volume of 733.33 cm^3^, with statistically significant differences compared with the 10% (449.17 cm^3^) and 20% (465.00 cm^3^) added bread. Storage caused a reduction in maximum volume in the control of 3.4% at T2 and 20.5% at T4, while the minimum in INU20 of 0.9% at T2 and 4.6% at T4.

Height followed the same trend as volume, with maximum values in the control (9.23 cm) and minimum values in INU10 (6.54 cm) at T0, reducing to values of 8.00 cm in Flour 0 and 6.00 cm in INU10 at T4.

The decrease in volume and height with increasing addition of inulin is typical of all fibrous baked products. As other authors [46,47,48] have also found, the addition of fiber (soluble or insoluble) leads to a deterioration of the gluten and a lower capacity to retain gas during fermentation.

Regarding porosity, the addition of inulin influenced the viscoelastic properties [49] by reducing the number of pores, as also observed by other authors. The reduction in porosity is directly proportional to the increase in the percentage of inulin [50]. The values remain constant for all storage times.

TPA analysis showed statistically significant differences between the control bread and those supplemented with inulin (Table 4).

The hardness was evaluated and revealed the lowest values for INU5 (4.19 N) and highest values for INU20 (12.63 N) at T0. During storage, the control bread increased its hardness to a greater extent both at T2 (17.67 N) and at T4 (22.96 N), with lower values for the inulin-containing bread (INU10, 13.37 N at T2 and INU5, 20.20 N at T4). The values found are in agreement with those reported by other authors [51] who used supplements up to 10% inulin, obtaining values for 5% (3.64 N) and 10% (10.23 N), similar to those described in the following work.

The elasticity values did not show statistically significant differences, with mean values close to each other around 0.95 at T0. This was different for the measurements during storage, with lower values due to the staling process, ranging from 0.76 (INU20) to 0.94 (INU10) at T2 and between 0.75 (Flour 0) and 0.95 (INU10) at T4.

Cohesion followed the same trend as elasticity over the different days of storage. The maximum value of 0.90 was measured in the control group and in INU5. With the increasing amount of inulin, the value decreased to 0.76 in INU20 at T0. At T2, mean values of 0.70 were found, and no statistically significant difference was found. At T4, a worsening and a substantial reduction in cohesion occurred, especially in the control bread (0.34), with higher values in INU10 (0.78).

The data reported for both elasticity and cohesion are consistent with what has been described by other authors [51].

Gumminess and chewiness parameters were strongly influenced by increased inulin supplementation. It was observed that at T0, the control had mean values of 3.13 N (gumminess) and 3.01 N × mm (chewiness), while INU10 had higher values of 15.81 N (gumminess) and 15.23 N × mm (chewiness). Storage had a strong worsening effect, with an increase observed at T2 for both gumminess (control 11.63 N and INU20 41.74 N) and chewiness (INU5 5.09 N × mm and INU10 36.60 N × mm). The same behavior was observed in T4, where the minimum gumminess was 21.26 N (control) and the maximum 200.59 N (INU10), while the chewiness ranged from 15.01 N × mm in the control to 190.44 N × mm in INU10.

The increase in bread hardness and the deterioration of gumminess and chewiness properties may depend not only on the physiological deterioration of bread during storage but also on the lower moisture content of inulin-enriched bread [52].

In general, the characteristics of TPA depended on the content of gluten-forming proteins [53]. The addition of inulin to wheat flour reduced its strength, causing less growth during fermentation, resulting in less volume and porosity.

The color analysis performed on the bread highlights what has already been reported by other authors [44,51]. As observed by Hager et al. [54], the addition of inulin caused browning of the crust of the loaves supplemented with inulin. This is due to the partial degradation of polysaccharides and monosaccharides present in the inulin powder, which is due to a stronger Maillard reaction, which is a nonenzymatic reaction between amino acids and reduced sugars at temperatures above 110 °C and generates the described effect. As reported in Table 5, the brown crust index tends to increase in direct proportion to the increase in inulin, rising from 29.45 in the control to 51.67 in the INU20 sample. Conversely, in the crumb, an increase in inulin corresponds to a decrease in the brown index, which characterizes very light crumbs (from 38.39 in the control to 23.47 in the INU20).

The red and yellow indices in the control crust (a* 4.00 and b* 26.77) were also lower than those found in the fortified bread, with maximum values of a* in INU20 (16.48) and b* in INU10 (33.97).

Even for the crumb, the red and yellow indices of the control bread were lower than those of the fortified breads. The range for a* was between −1.27 (control) and 0.05 (INU20), and for b*, between 11.77 (control) and 16.68 (INU10). The differences observed in the individual color coordinates, summarized in ΔE, confirm what was previously indicated. The use of inulin produces progressively greater color differences compared with the control bread for both the crust and the crumb.

### 3.4. Nutritional Characteristics of the Breads

Given the importance of inulin enrichment in wheat bread for its prebiotic role, in this study its degradation during the baking procedure was also evaluated. As shown in Table 6, the inulin content in the bread samples, quantified as fructose through acid hydrolysis, was lower than the amount added during bread making. The amount of inulin, ranging from approximately 0.2 to 14.4 g/100 g, was found to be proportional to the added content and statistically different (*p* < 0.05). On average, about 33% of the inulin was lost, but the INU20 bread, containing 14.42 g of inulin per 100 g of sample, was the sample with the lowest loss, at 28%.

As reported by several authors, the leavening and the baking process cause a reduction in inulin content ranging from 25 to 41% because its stability could be influenced by the degree of polymerization (DP) and the type of fermentation [55,56]. Specifically, the baking process may have led to the hydrolysis of low molecular weight fructans, increasing the levels of free sugars, especially fructose, which promote the Maillard reaction. Furthermore, the polymerization degree of inulin could also influence its degradation; in fact, low DP inulin is degraded more easily during the baking process. Additionally, the yeast *Saccharomyces cerevisiae* produces invertase, an enzyme capable of breaking down low molecular weight fructans, making them more susceptible to hydrolysis during cooking [57,58]. Previous studies have shown that the prebiotic effects of inulin are determined with a daily intake of approximately 5 g of it, which induces beneficial effects on the intestinal microbiota level [55,59]. For this reason, considering a 50 g portion of the bread, the consumption of INU20 is the most suitable to achieve this target.

Based on in vitro analysis, the effect of inulin enrichment on the glycemic index (GI) of breads was evaluated. The hydrolysis index (HI) was calculated by comparing the area under the hydrolysis curve of the bread samples with that of white wheat bread (HI = 100), which is used as the standard food. The HI proved to be an effective predictor of glycemic response (pGI) after food consumption and showed a strong correlation with GI in vivo. As reported in Table 6, the presence of inulin showed an effect on HI and, consequently, a reduction of pGI. The data demonstrated a similar elevated pGI (*p* > 0.05) to that of wheat white bread for Flour 0 (94.52 ± 0.76) and INU5 (94.20 ± 0.74) breads and a statistically different reduction (*p* < 0.05) for INU10 and INU20 breads. Accordingly, the pGI value of INU20 bread, corresponding to 89.39 ± 0.28, was the lowest (Table 6). These results are in agreement with those of other studies that have shown a reduction in pGI proportional to the percentage of inulin added to the formulation of different bread products [55,60].

### 3.5. Principal Component Analysis

The principal component analysis (Figure 5) resulted in the first two components representing 92.5% of the total explained variance (PC1 = 64.8%; PC2 = 27.7%), with the four theses under study clearly distinguished in the multidimensional space (Appendix A).

INU10 and INU20 were represented in the positive section of the Principal Component 1 (Appendix A), which was highly positively correlated with mixing time, dough development, P/L, crumb brown index and a* of the crumb, as well as with weight and porosity at T0, T2 and T4. Additionally, a strong positive correlation was noted with gumminess and chewiness at T0 and T2, and cohesiveness at T4.

On the other hand, INU10 and INU20 were highly negatively correlated with WB, water absorption, the brown index of the crust, as well as with the volume, height and moisture of the breads collected at T0, T2 and T4, hardness at T4 and cohesiveness at T0.

Flour 0, due to its high negative score, together with INU5 to a lesser extent, both being positioned in the negative part of PC1, had characteristics opposite to those of INU10 and INU20.

INU5 and INU10, positioned in the positive part of PC2, were also positively correlated with OB, stability, b* of the crust, springiness at T0, T2 and T4, gumminess and chewiness at T4 and cohesiveness at T2, in addition to being negatively correlated with hardness at T0 and T2 (Figure 6).

## 4. Conclusions

The aim of this paper was to evaluate in broader terms the impact of inulin on the technological and nutritional properties of fortified bread, specifically its direct effect on lowering the glycemic index.

From a technological point of view, the reduction of water absorption with increasing inulin addition represents a major problem in this study. The decrease in water absorption due to the higher integration of inulin contributes negatively to the usual dough formation and fermentation activities of leavened bakery products.

The farinograph test showed that at 5% of inulin integration, the dough development time greatly increased while the stability decreased compared with the control. The reduction in water absorption capacity, together with higher soluble fiber content, increased the strength (W) and toughness (P/L) of the doughs, as measured by the alveograph.

Regarding color, the activation of the Maillard reaction in the loaves with inulin produced a darker crust as the supplementation increased, while in the crumb the presence of inulin resulted in a whiter color compared with the control.

The physical-chemical analyses on doughs and rheological analyses on the breads have shown that supplementation with increasing amounts of inulin can negatively influence the technological properties of the finished products. The 20% integration caused a marked reduction in water absorption in the dough, with consequent worsening of the texture profile characteristics.

On the other hand, increasing the percentage of inulin in breads improved the nutritional profile related to the glycemic index in vitro.

In conclusion, high inulin supplements are of interest to produce baked products with a lower glycemic index and high fiber content, but, on a technological level, additions above 5% inulin are not recommended. The sample supplemented with 5% inulin gave the best results. However, it should be noted that the range, including 10–20% supplementation, is interesting for further work from a sensory point of view.

## Figures and Tables

**Figure 1 foods-13-02711-f001:**
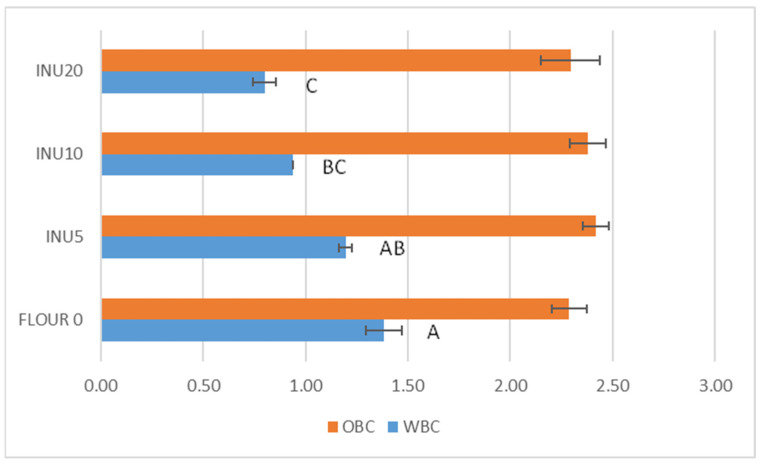
Water-binding capacity (WBC; g H_2_O water/g d.m. flour) and oil-binding capacity (OBC; g oil/g d.m. flour) of the samples of flours with integration of inulin (0, 5, 10, 20% from bottom to top). Different letters in column indicate a significant difference: *p* ≤ 0.01 (Tukey) for WBC; *p* = n.s. (Tukey) for OBC.

**Figure 2 foods-13-02711-f002:**
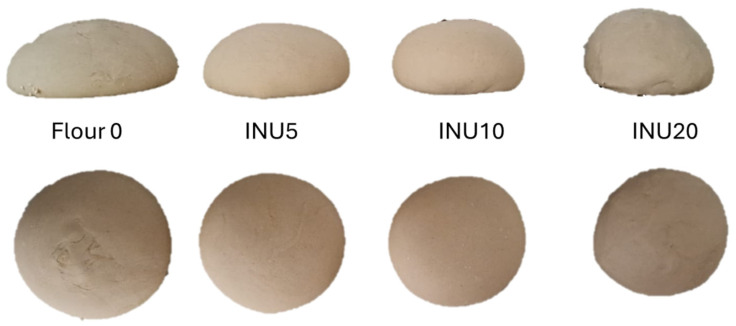
Appearance of 250 g doughs produced with integration of inulin (0, 5, 10, 20% from left to right).

**Figure 3 foods-13-02711-f003:**
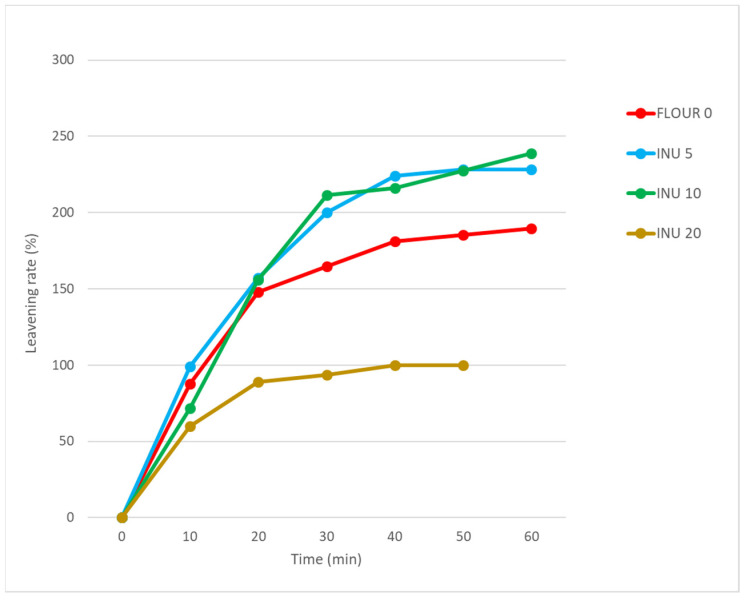
Leavening rate (%) at increasing levels of replacement (0, 5, 10, 20%) of the samples of flours with integration of inulin.

**Figure 4 foods-13-02711-f004:**
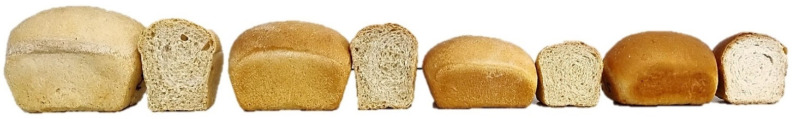
Bread with increasing levels of integration (0, 5, 10, 20%) of the flour samples with inulin integration.

**Figure 5 foods-13-02711-f005:**
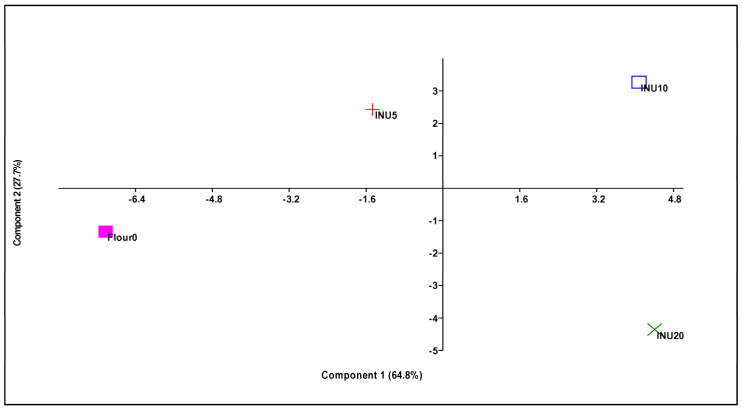
Principal component analysis (PCA) scatter diagram defined by the first two principal components.

**Figure 6 foods-13-02711-f006:**
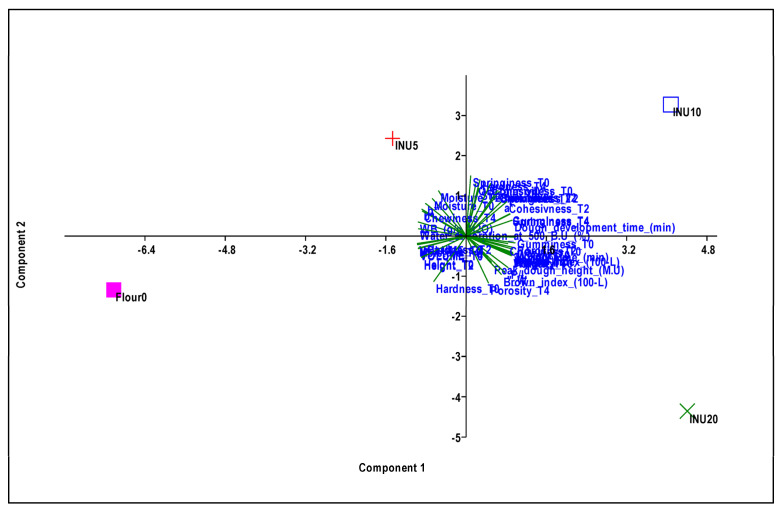
Principal component analysis (PCA) biplot. Vectors represent the loadings of the physical and chemical characteristics of flours, doughs, and breads with different inulin integration.

**Table 1 foods-13-02711-t001:** Formulation of the experimental bread (g/100 g of wheat flour).

Bread Type	Flour (g)	Commercial Inulin (g)	Commercial Yeast (g)	NaCl (g)	Water (g)
Flour 0	100	-	1.5	2.2	58
INU 5	95	5	1.5	2.2	48
INU 10	90	10	1.5	2.2	45
INU 20	80	20	1.5	2.2	41

**Table 2 foods-13-02711-t002:** Rheological data of the samples of flours with integration of different inulin.

Sample	Farinograph			Mixograph		Alveograph	
	Dough Development Time (min)	Stability (min)	Water Absorption at 500 B.U. * (%)	Mixing Time (min)	Peak Dough Height (M.U.) **	W (10^−4^ × J)	P/L
Flour 0	1.80 ± 0.28 b	8.15 ± 0.35 b	57.50 ± 0.28 a	4.61 ± 0.08 d	5.78 ± 0.11	154 ± 8.49 b	0.82 ± 0.02 c
INU5	14.80 ± 0.85 a	8.20 ± 0.42 b	48.25 ± 0.21 b	6.90 ± 0.37 c	5.48 ± 0.04	150 ± 2.83 b	2.45 ± 0.17 bc
INU10	18.35 ± 0.07 a	13.10 ± 0.28 a	44.85 ± 1.06 bc	7.32 ± 0.02 b	5.93 ± 0.11	175 ± 2.83 ab	3.21 ± 0.20 ab
INU20	16.65 ± 0.07 a	6.35 ± 0.21 b	40.75 ± 0.21 c	8.70 ± 0.04 a	5.95 ± 0.35	203 ± 1.41 a	4.55 ± 0.04 a

Different letters in column indicate a significant difference: *p* < 0.001 for dough development time, farinograph stability, water absorption and mixing time at mixograph, P/L alveographic; *p* < 0.01 for W alveographic. * Brabender Unit. ** Mixograph Unit.

**Table 3 foods-13-02711-t003:** Change in physical characteristics of bread with increasing level of inulin integration (0, 5, 10, 20%) during 5 days of storage.

Days	Type of Bread	Moisture (g/100 g)	ΔRH (%)	Weight (g)	Volume (cm^3^)	Height (cm)	Porosity (1–8) *
T0	Flour 0	35.72 ± 0.01 a	-	153.99 ± 0.89 c	773.33 ± 10.41 a	9.23 ± 0.06 a	4.33 ± 0.29 b
	INU5	33.50 ± 0.01 a	-	156.70 ± 0.76 ab	574.17 ± 10.10 b	8.39 ± 0.03 ab	4.50 ± 0.25 b
	INU10	32.15 ± 0.01 ab	-	159.44 ± 0.74 ab	449.17 ± 1.44 c	6.54 ± 0.25 c	7.07 ± 0.38 a
	INU20	27.06 ± 0.01 b	-	160.92 ± 1.81 a	465.00 ± 9.01 c	7.32 ± 0.17 b	7.53 ± 0.35 a
T2	Flour 0	31.67 ± 0.02 a	11.3	142.70 ± 1.09 c	746.67 ± 20.21 a	8.35 ± 0.06 a	4.33 ± 0.29 b
	INU5	29.81 ± 0.01 ab	11.0	148.01 ± 0.75 bc	541.67 ± 12.58 b	7.71 ± 0.03 ab	4.50 ± 0.25 b
	INU10	29.87 ± 0.00 ab	7.1	153.02 ± 0.40 ab	449.17 ± 6.61 c	6.33 ± 0.25 c	7.07 ± 0.38 a
	INU20	25.73 ± 0.01 b	4.9	156.19 ± 1.86 a	460.83 ± 9.46 c	7.20 ± 0.17 b	7.53 ± 0.35 a
T4	Flour 0	27.80 ± 0.01 a	22.2	131.32 ± 1.09 c	615.00 ± 7.07 a	8.00 ± 0.14 a	5.13 ± 0.18 c
	INU5	27.21 ± 0.01 a	18.8	141.29 ± 0.57 bc	520.50 ± 7.78 ab	7.53 ± 0.15 ab	6.13 ± 0.18 bc
	INU10	27.05 ± 0.00 ab	15.9	144.79 ± 0.47 ab	422.50 ± 3.54 ab	6.00 ± 0.10 c	7.38 ± 0.18 b
	INU20	24.32 ± 0.00 b	10.1	153.20 ± 1.46 a	443.75 ± 8.84 b	6.77 ± 0.32 b	7.53 ± 0.35 a

* Scale 1–8; 1 = non-uniform structure, large and irregular cells; 8 = uniform compact structure, small and regular cells. Different lower-case letters in a column indicate a significant difference *p* ≤ 0.001, *p* ≤ 0.01 (weight at T2) and *p* ≤ 0.05 (moisture at T4) for the same type of bread at different storage times.

**Table 4 foods-13-02711-t004:** Texture profile analysis (TPA) of bread with increasing levels of inulin supplementation (0, 5, 10, 20%) during 5 days of storage.

Days after Baking	Type of Bread	Hardness (N)	Springiness	Gumminess (N)	Chewiness (N × mm)	Cohesiveness
T0	Flour 0	8.59 ± 0.45 b	0.95 ± 0.01	3.13 ± 0.02 d	3.01 ± 0.05 d	0.90 ± 0.01 a
	INU5	4.19 ± 0.88 c	0.95 ± 0.02	7.25 ± 0.01 c	6.97 ± 0.08 c	0.90 ± 0.01 a
	INU10	8.66 ± 0.53 b	0.96 ± 0.01	15.81 ± 0.03 a	15.23 ± 0.19 a	0.81 ± 0.04 ab
	INU20	12.63 ± 0.90 a	0.94 ± 0.02	10.30 ± 0.16 b	9.78 ± 0.08 b	0.76 ± 0.05 b
T2	Flour 0	17.67 ± 0.06 a	0.81 ± 0.01 b	11.63 ± 0.01 d	9.50 ± 0.13 c	0.65 ± 0.05 a
	INU5	14.70 ± 1.55 ab	0.90 ± 0.02 a	27.76 ± 0.11 c	5.09 ± 0.07 d	0.69 ± 0.03 a
	INU10	13.37 ± 1.08 b	0.94 ± 0.03 a	39.00 ± 0.17 b	36.60 ± 0.05 a	0.71 ± 0.02 a
	INU20	16.17 ± 1.87 ab	0.76 ± 0.01 b	41.74 ± 0.17 a	32.17 ± 0.04	0.64 ± 0.01 a
T4	Flour 0	22.96 ± 3.09 a	0.75 ± 0.01 c	21.26 ± 0.47 d	15.01 ± 1.03 c	0.34 ± 0.01 c
	INU5	20.20 ± 0.87 b	0.86 ± 0.01 b	104.14 ± 1.32 b	85.18 ± 5.40 b	0.64 ± 0.02 ab
	INU10	21.87 ± 0.70 ab	0.95 ± 0.01 a	200.59 ± 2.81 a	190.44 ± 2.93 a	0.78 ± 0.01 a
	INU20	21.34 ± 2.02 b	0.81 ± 0.01 bc	62.15 ± 2.84 c	49.88 ± 2.36 bc	0.58 ± 0.01 b

Different lower-case letters in a column indicate a significant difference *p* ≤ 0.001 (hardness at T0, gumminess and chewiness at T0 and T2); *p* ≤ 0.01 (cohesiveness at T0; hardness at T4); *p* ≤ 0.05 (hardness and cohesiveness at T2) for the same type of bread at different storage times.

**Table 5 foods-13-02711-t005:** Colorimetric parameters of bread crust and crumb at increasing levels of substitution (0, 5, 10, 20%) of flour with inulin integration.

Sample	Crust				Crumb			
	Brown Index (100 − L*)	a*	b*	ΔE	Brown index	a*	b*	ΔE
Flour 0	29.45 ± 0.43 c	4.00 ± 1.05 d	26.77 ± 2.40 b	0.0	38.39 ± 1.75 a	−1.27 ± 0.03 b	11.77 ± 1.14 c	0.0
INU5	31.34 ± 1.70 bc	8.33 ± 0.61 c	32.40 ± 0.92 ab	27.0	35.29 ± 0.79 ab	−1.00 ± 0.20 b	15.40 ± 0.35 ab	11.4
INU10	39.92 ± 2.36 b	12.36 ± 0.25 b	33.97 ± 1.53 a	115.7	30.98 ± 0.69 b	−0.91 ± 0.06 b	16.68 ± 0.11 a	39.6
INU20	51.67 ± 2.31 a	16.48 ± 0.78 a	30.13 ± 1.73 ab	330.4	23.47 ± 0.89 c	0.05 ± 0.06 a	13.05 ± 0.27 bc	113.0

Different letters in column indicate a significant difference: *p* ≤ 0.001 (Tukey); *p* ≤ 0.01 (Tukey) for b* crust.

**Table 6 foods-13-02711-t006:** Changes in inulin content and predicted glycemic index of the bread at increasing levels of replacement (0, 5, 10, 20%) of flours with inulin integration.

	Bread			
Sample	Flour 0	INU5	INU10	INU20
Inulin (g/100 g)	0.18 ± 0.02 d	3.53 ± 0.160 c	6.32 ± 0.21 b	14.42 ± 0.07 a
Hydrolysis Index (HI)	99.83 ± 1.39 a	99.26 ± 1.34 a	95.48 ± 1.49 b	90.48 ± 0.52 c
Predicted Glycemic Index (pGI)	94.52 ± 0.76 a	94.20 ± 0.74 a	92.13 ± 0.82 b	89.39 ± 0.28 c

The data are the means of three independent experiments ± standard deviations (*n* = 3). a–d Values in the same row with different letters differ significantly (*p* < 0.05).

## Data Availability

The original contributions presented in the study are included in the article/Appendix A, further inquiries can be directed to the corresponding author.

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
