# Peer review of "Technological Properties of Inulin-Enriched Doughs and Breads, Influence on Short-Term Storage and Glycemic Response"

_foods, 2024, doi:10.3390/foods13172711_

Round 1
Reviewer 1 Report
Comments and Suggestions for Authors
This paper deals with the effect of inulin addition on the physical, technological and rheological properties of the flour mixtures. This is a very interesting study for the promotion of inulin in bread industry. The data is solid and the results are useful to the readers and the consumers.
Several questions,
1) Line 70-88, although inulin is prebiotic fiber, inulin is prohibited to eat by infants and pre-school children. Moreover, a recent report (Singh, et al 2018) showed that dysregulated microbial fermentation of inulin can induce cholestatic liver cancer. V. Singh, B.S. Yeoh, B. Chassaing, et al., Dysregulated microbial fermentation of soluble fiber induces cholestatic liver cancer, Cell 175 (2018)679–694. The authors had better give a objective guidance for readers.
2) Line 144, only a word was used to introduce the material of inulin. Now inulin has been produced by many countries in despite of different processing technology, the characteristics of Italian inulin should be introduced, including average molecular weight, degree of polymerization, purity, and so on. The used inulin was from chicory or Jerusalem artichoke?
3) The abstract is too long, and the detailed experimental results were given, but the definite conclusion should be given. How many grams of inulin are suitable for adding to the dough during the bread-making?
4) Line 22, The theses with 5 and 10% inulin supplementation, please check “the theses”.
5) In the conclusion section, some conclusions were not derived from the present study.
6) The principal component analysis had better be used in this study for a great number of experimental data.
Author Response
This paper deals with the effect of inulin addition on the physical, technological and rheological properties of the flour mixtures. This is a very interesting study for the promotion of inulin in bread industry. The data is solid and the results are useful to the readers and the consumers.
Several questions,
1) Line 70-88, although inulin is prebiotic fiber, inulin is prohibited to eat by infants and pre-school children. Moreover, a recent report (Singh, et al 2018) showed that dysregulated microbial fermentation of inulin can induce cholestatic liver cancer. V. Singh, B.S. Yeoh, B. Chassaing, et al., Dysregulated microbial fermentation of soluble fiber induces cholestatic liver cancer, Cell 175 (2018)679–694. The authors had better give a objective guidance for readers.
We thank the reviewer for the important observation. We have added, as correctly suggested, the negative effects associated with inulin intake in the diet of infants and pre-school children (lines 63-68).
2) Line 144, only a word was used to introduce the material of inulin. Now inulin has been produced by many countries in despite of different processing technology, the characteristics of Italian inulin should be introduced, including average molecular weight, degree of polymerization, purity, and so on. The used inulin was from chicory or Jerusalem artichoke?
As requested by the reviewer, we have reported additional information on inulin. The inulin used is not of Italian origin (only the retailer is). Regarding the main chemical characteristics, what is indicated in the manufacturer's technical sheet (lines 106 - 116) has been reported.
It is a short-chain inulin with a purity level of 90%. It comes from the chicory root.
3) The abstract is too long, and the detailed experimental results were given, but the definite conclusion should be given. How many grams of inulin are suitable for adding to the dough during the bread-making?
We thank the reviewer for this observation that allowed us to improve the abstract. In particular, we shortened the abstract by removing some detailed experimental results and added the final conclusion. Furthermore, we reported the sample that showed the best results (formulation with 5% inulin).
4) Line 22, The theses with 5 and 10% inulin supplementation, please check “the theses”.
We thank the reviewer for this comment. We have replaced 'the thesis' with 'samples'.
5) In the conclusion section, some conclusions were not derived from the present study.
We welcomed the reviewer's suggestion and revised the conclusions paragraph and eliminated conclusions not related to this study.
6) The principal component analysis had better be used in this study for a great number of experimental data.
We thank the reviewer for this suggestion of multivariate statistics that has significantly improved the interpretation of the results. In fact, we have inserted the relevant part in the Materials and Methods paragraph (lines 266-269) and a new paragraph (3.6 Principal Components Analysis).

Reviewer 2 Report
Comments and Suggestions for Authors
Dear Authors,
The manuscript with title "Technological properties of inulin-enriched doughs and breads, influence on short-term storage and glycemic response" is interesting and deals with the overall quality of bread produced by different amount of inline as a very important prebiotic for human health.
However, I will suggest improving of the manuscript in several points:
1. Which type of inulin was used long chain inulin or short chain inulin? This is very important because protective effect of long and short chain inulin is not the same. 2.5% long‐chain inulin had a greater protective effect on gluten protein compared with that containing 5% short‐chain inulin during the production of bread.
2. Which was the pH of the prepared experimental dough on Table 1? This is very important due to the fact that yeast‐based bread has a pH of about 5–6, while sourdough bread has about 4 because of the lactic acid bacteria it contains.
3. Baking process should be separately discussed because leads to the destruction of 20%–100% of fructan chains and the production of new low‐molecular‐weight products such Di‐D‐fructose dianhydrides.
4. The authors should discuss the effect of inulin of the overall taste and smell of produced breads in terms of Millard reaction, which is a nonenzymatic reaction between amino acids and reducing sugars at temperatures above 110°C.
I suggest major revision of the manuscript.
Comments on the Quality of English Language
Dear Editors,
The manuscript with title "Technological properties of inulin-enriched doughs and breads, influence on short-term storage and glycemic response" is interesting and deals with the overall quality of bread produced by different amount of inline as a very important prebiotic for human health.
However, I will suggest improving of the manuscript in several points:
1. Which type of inulin was used long chain inulin or short chain inulin? This is very important because protective effect of long and short chain inulin is not the same. 2.5% long‐chain inulin had a greater protective effect on gluten protein compared with that containing 5% short‐chain inulin during the production of bread.
2. Which was the pH of the prepared experimental dough on Table 1? This is very important due to the fact that yeast‐based bread has a pH of about 5–6, while sourdough bread has about 4 because of the lactic acid bacteria it contains.
3. Baking process should be separately discussed because leads to the destruction of 20%–100% of fructan chains and the production of new low‐molecular‐weight products such Di‐D‐fructose dianhydrides.
4. The authors should discuss the effect of inulin of the overall taste and smell of produced breads in terms of Millard reaction, which is a nonenzymatic reaction between amino acids and reducing sugars at temperatures above 110°C.
I suggest major revision of the manuscript.
Author Response
Dear Authors,
The manuscript with title "Technological properties of inulin-enriched doughs and breads, influence on short-term storage and glycemic response" is interesting and deals with the overall quality of bread produced by different amount of inline as a very important prebiotic for human health.
However, I will suggest improving of the manuscript in several points:
- Which type of inulin was used long chain inulin or short chain inulin? This is very important because protective effect of long and short chain inulin is not the same. 2.5% long‐chain inulin had a greater protective effect on gluten protein compared with that containing 5% short‐chain inulin during the production of bread.
We thank the Reviewer for the question as it allows us to provide important information about the type of inulin used. We used short-chain inulin. In order to answer another reviewer, we have reported additional information as indicated by the manufacturer's technical sheet (lines 106 – 116).
- Which was the pH of the prepared experimental dough on Table 1? This is very important due to the fact that yeast‐based bread has a pH of about 5–6, while sourdough bread has about 4 because of the lactic acid bacteria it contains.
The pH of our doughs ranged between 5.6 and 5.7. As indicated in the M&Ms section (lines 151-152), commercial yeast (predominant Saccharomyces cerevisiae) was used, not sourdough (in which lactic acid bacteria are present). For further clarity, we added the word ‘commercial’ to line 153 and in the descriptor in Table 1. As is known, Saccharomyces cerevisiae tends to dominate the entire fermentation process at the expense of potential, if accidentally present, lactic acid bacteria, so the pH, as also reported by the reviewer, remains between 5 and 6.
- Baking process should be separately discussed because leads to the destruction of 20%–100% of fructan chains and the production of new low‐molecular‐weight products such Di‐D‐fructose dianhydrides.
The aim of this work is to evaluate the influence of inulin from a food and nutritional technology point of view only in relation to in vitro glycemic index assessment. In this sense, the described cooking process falls within the scope of the work. Chemical investigations of simple sugar conversion processes were not carried out. These relate to another topic, which is different from the subject of this study, as indicated in the title, abstract and conclusions.
- The authors should discuss the effect of inulin of the overall taste and smell of produced breads in terms of Millard reaction, which is a nonenzymatic reaction between amino acids and reducing sugars at temperatures above 110°C.
We agree with the Reviewer that a sensory evaluation would add value to the study, but at the moment we have not the possibility to perform such analysis. However, we are taking into account this suggestion and the sensory properties of the breads with added of inulin will be deepened in a future study. This study, indeed, has to be intended as a first work to establish the rheological and technological conditions for an optimal product development, in order to identify the optimal percentage of inulin to be used. In collaboration with other teams and as indicated in the conclusions (lines 536-540) the sensory analysis will be carried out in subsequent studies with inulin concentrations from 10 to 20% both in pure form and in mixture with other matrices. We discussed the Maillard reaction in lines 445-464, where we revised the discussion and added the specification given by the Reviewer.
